# Instruction-Level Weight Shaping: A Framework for Self-Improving AI Agents

## Abstract

Large language models (LLMs) excel at surface fluency yet remain structurally static after pre-training; new or evolving domain knowledge is typically bolted on via retrieval-augmented generation (RAG) or parameter fine-tuning, but RAG often retrieves facts without integrating them logically and adds latency, while fine-tuning is resource-intensive and risks catastrophic forgetting. We propose Instruction-Level Weight Shaping (ILWS), which treats curated system instructions as external, auditable pseudo-parameters updated post-session via reflection and user feedback: after each session an LLM-driven Reflection Engine inspects the conversation trace, diagnoses reasoning successes or failures, and proposes typed deltas $\Delta K = (\Delta S, \Delta U, \Delta T)$ over instructions, user preferences, and tools; each delta is version-controlled, evaluated under a sliding-window analysis of 1–5 star ratings, automatically repaired on first failure, and rolled back on repeated failure; and when the accumulated edit budget crosses a threshold, the agent can optionally compile a rating-weighted synthetic dataset and distil matured instruction-space gains into parameters. Empirically, ILWS makes explicit the low-rank shaping implicitly induced by context in transformer blocks and preserves governance while eliminating per-call retrieval: in a real-world e-commerce platform proof of concept (PoC) called "L0 Support" with 1M-token context, a single operator using the reflection-driven knowledge accumulation achieved 4–5× gains in tickets/hour and ∼80% reduction in time per ticket, with first-shot resolution improving from ∼20% to ∼90%; when the matured instruction base was deployed to six additional operators without further reflection updates, they reported comparable gains, suggesting that ILWS produces transferable domain specialisation akin to fine-tuning but without parameter modification. Because ILWS operates at the instruction layer, it generalises to dynamic domains (legal, medical, engineering) requiring adaptive reasoning, tool creation, and low-latency deployment.

## 1 Introduction

This paper introduces Instruction-Level Weight Shaping (ILWS), a lightweight framework for continual self-improvement in large language models (LLMs). In production systems, system instructions are treated as authoritative directives. ILWS reinterprets these instructions not as fixed configuration, but as a mutable, externalised memory channel, a low-cost, auditable surrogate for the model's internal weights. Rather than updating parameters through costly fine-tuning or repeatedly fetching context via retrieval-augmented generation (RAG), ILWS uses a post-session reflection and feedback loop to produce knowledge deltas. These fine-grained edits gradually evolve the system prompt to better capture a domain's logic, tools, and user expectations.

We contend that a large fraction of *operational* evolution can be handled by structured edits to the system prompt itself, provided edits are (i) feedback-driven and quantitative, (ii) reversible under governance, and (iii) recorded with code-like rigour. After each session, a stochastic Reflection Engine proposes a knowledge delta $\Delta K_t = (\Delta S_t, \Delta U_t, \Delta T_t)$, which is trialled, score-gated (accepted only if a sliding-window rating improves by at least $\tau$ with significance $\alpha$), possibly repaired, and either accepted into a versioned knowledge

state or rolled back. When the cumulative instruction budget exceeds a threshold, ILWS synthesises a rating-weighted dataset and distils persistent instruction-space edits into model weights (cf. Optional distillation, Eq. (5)).

Formally, at turn $t$ the agent uses a frozen backbone $f_\theta$ and a composite knowledge state $K_t = (S_t, U_t, T_t)$:

$$\hat{y}_t = f_\theta(x_t, K_t), \qquad K_{t+1} = (S_t \oplus \Delta S_t, \ U_t \oplus \Delta U_t, \ T_t \oplus \Delta T_t). \tag{1}$$

Here, $\oplus$ denotes applying the typed edit list to each component (insert/modify/delete). We show how this operational loop provides an explicit, auditable analogue of *implicit* low-rank weight shaping induced by prompts in transformer blocks (akin to low-rank adapters (LoRA) produced on-the-fly by prompts (Hu et al., 2022; Li et al., 2022)), and how it complements RAG and fine-tuning in practice.

**Contributions.**

- **Formalise ILWS with typed deltas and governance.** We cast structured prompt edits over $K = (S, U, T)$ as explicit, version-controlled surrogates for low-rank weight updates, with git-backed persistence and rollback.

- **Statistical gate with repair/rollback.** We introduce a sliding-window, statistically grounded acceptance rule parameterised by $(\tau, \alpha)$, admitting edits only if average ratings improve by at least $\tau$ with significance $\alpha$, with one-shot repair and rollback on second failure.

- **Theory link to implicit low-rank shaping.** We articulate how instruction edits influence prompt-conditioned activations to yield LoRA/IA$^3$-like low-rank effects, providing an explicit, auditable analogue of implicit updates induced by prompts.

- **Production architecture with autonomous tool synthesis.** We present a stochastic reflection loop, sandboxed tool generation and integration, and operational guardrails (policy invariants, statistical change detection on ratings).

- **Empirical deployments and latency/cost profile.** (i) Three-month SRE study with 2.4–5.0× throughput gains and ∼80% fewer audited hallucinations; (ii) E-commerce platform PoC "L0 Support" with 4–5× tickets/hour and ∼80% time-per-ticket reduction. ILWS adds no per-call retrieval; RAG incurred +300–600 ms median and up to +2000 ms at the 95th percentile (p95) per turn in our measurements.

## 2 Theoretical context and related work

**Implicit low-rank updates from context.** A transformer block comprising a contextual layer followed by a multi-layer perceptron (MLP) can be viewed as applying an *implicit* rank-one update to the first MLP layer, computed from the context tokens; iteratively consuming tokens yields an online gradient-like dynamic over an effective weight matrix (Dherin et al., 2025). ILWS makes such influences *explicit* by editing instructions outside the network, preserving auditability and persistence.

**Path-kernel view of trained models.** Models trained by gradient descent are approximately kernel machines whose predictions can be expressed via *path kernels* that integrate gradient similarities along the optimization trajectory (Domingos, 2020). Under this view, retrieved or prompted examples act as non-parametric supports. Selecting context that aligns with gradient-path features should improve reliability, a perspective we exploit in our retrieval-free but *instruction-grounded* edits.

**RAG and modular augmentation.** RAG is effective but requires careful indexing, retrieval granularity, reranking, compression, and adaptive triggering to avoid irrelevant or counterfactual context (Gao et al., 2024). ILWS sidesteps per-call retrieval by baking vetted rules and patterns into instructions, while remaining compatible with optional RAG when needed.

**Self-improvement and reflection.** Memory-augmented prompting (e.g., MemPrompt (Madaan et al., 2022)), reflection-based retries (e.g., Reflexion (Shinn et al., 2023)), and active retrieval (e.g., Self-RAG, FLARE (Asai et al., 2023; Jiang et al., 2023)) demonstrate gains from critique and adaptation. ILWS unifies these ideas *post-session*, gating durable edits with human-centric scores and governance, and adding autonomous tool synthesis.

**LLM tool-use and modular retrieval.** Orthogonal lines of work teach models to call tools (e.g., Toolformer) and to interleave retrieval and generation (e.g., Demonstrate-Search-Predict, active retrieval) (Schick et al., 2023; Khattab et al., 2022; Jiang et al., 2023; Asai et al., 2023). These systems improve evidence acquisition at inference time, but typically do not persist successful reasoning as durable edits. ILWS is complementary: it converts recurrent patterns into authoritative instructions and (optionally) new tools, reducing online retrieval while preserving auditability.

## 3 Problem formulation

Let $f_\theta$ be a frozen LLM backbone. The agent maintains a *knowledge state*

$$K_t = (S_t, U_t, T_t), \tag{2}$$

where $S_t$ is the current instruction set, $U_t$ captures user learnings or preferences, and $T_t$ is the registry of callable tools. During inference the system prompt is constructed deterministically from $K_t$ and provided to $f_\theta$ together with the user input $x_t$; cf. Eq. (1). At session end the Reflection Engine emits a candidate $\Delta K_t = (\Delta S_t, \Delta U_t, \Delta T_t)$.

**Score-gated acceptance.** Let $r_i \in \{1, \ldots, 5\}$ be user ratings (5-point Likert). A candidate edit $\Delta K_t$ is provisionally deployed immediately and evaluated over the next $N_{\text{win}}$ interactions. Define sliding-window means over $N_{\text{win}}$ interactions:

$$\bar{r}_{\text{prev}} = \frac{1}{N_{\text{win}}} \sum_{i=t-N_{\text{win}}}^{t-1} r_i, \qquad \bar{r}_{\text{new}} = \frac{1}{N_{\text{win}}} \sum_{i=t}^{t+N_{\text{win}}-1} r_i. \tag{3}$$

We accept the provisional edit if

$$\bar{r}_{\text{new}} \geq \bar{r}_{\text{prev}} + \tau \quad \text{and} \quad p\text{-value} \leq \alpha, \tag{4}$$

where the $p$-value comes from a one-sided Welch $t$-test by default (falling back to Mann–Whitney if normality fails Shapiro–Wilk), and typical hyper-parameters are $(\tau, \alpha) = (0.05, 0.05)$. Note that $\tau = 0.05$ corresponds to one-twentieth of a star on a 5-point scale. Equivalently, the gate admits an edit only if it improves the average rating by at least $\tau$ with significance $\alpha$. On first failure we solicit a typed automatic repair $\Delta K_t' = (\Delta S_t', \Delta U_t', \Delta T_t')$; on a second consecutive failure we roll back to the last tagged good state. We note that this gate is designed as an engineering safeguard for drift control, not as a formal hypothesis-testing framework; the sliding-window approach does not account for temporal autocorrelation or multiple testing across successive edits. In practice, the repair/rollback mechanism and optional human veto provide additional robustness beyond the statistical signal.

**Optional distillation.** We track a running instruction-change budget

$$B_T = \sum_{i=1}^{T} \left( \|\Delta S_i\| + \|\Delta U_i\| + \|\Delta T_i\| \right),$$

where $\| \cdot \|$ is a simple size metric such as token count or edit length. When $B_T \geq M$, we synthesise a rating-weighted dataset $D_{\text{syn}} = \{(x, K, \hat{y}, r)\}$ and solve

$$\theta^\star \in \arg\min_{\theta'} \sum_{(x,K,y) \in D_{\text{syn}}} w(x,y) \, \mathcal{L}_{\text{CE}}(f_{\theta'}(x,K), y) \quad \text{(token-level cross-entropy)}, \qquad w(x,y) = \frac{r-1}{4} \in [0,1], \tag{5}$$

then redeploy $f_{\theta^\star}$ and reset $B_T$.

# 4 Instruction-Level Weight Shaping (ILWS)

Figure 1 (left-to-right) depicts the four phases.

## 4.1 Phase 1: Inference

Given $(x_t, K_t)$, the agent returns $\hat{y}_t = f_\theta(x_t, K_t)$ per Eq. (1). In our reference system, $K_t$'s components are versioned, serialised JSON fragments deterministically composed into the system prompt; the schema is version-pinned so edits remain diff-friendly. The session stores a transcript, tool logs, and the rating $r_t$.

## 4.2 Phase 2: Post-session reflection and update

A session-end hook or cron job invokes the Reflection Engine $R$ with transcript, tool logs, and recent ratings. $R$ emits a *typed* delta

$$\Delta K_t = (\Delta S_t, \Delta U_t, \Delta T_t)$$

expressed as calls into a self-modification API (`appendInstruction`, `modifyInstruction`, `createTool`, `deprecateTool`, `addUserPreference`, . . . ) plus a structured rationale (YAML diagnostics, score deltas). The candidate is applied immediately and evaluated over the next $N_{\mathrm{win}}$ sessions under the score-gating rule (4).

**Autonomous tool synthesis.** If $\Delta T_t \neq \varnothing$, the Tool Manager compiles and unit-tests generated Python in a sandbox (networkless, no egress; OCI/seccomp profile). On success, the tool signature is appended to $T$, and a concise usage rubric is inserted into $S$ to make the capability discoverable by the model; admins may revert within a review window $\Delta t$.

## 4.3 Phase 3: Persistence and governance

Accepted deltas are committed to an immutable git repository and tagged with a knowledge checkpoint; admins act as observers and may optionally veto by triggering a one-click revert within a configurable review window $\Delta t$. A dashboard exposes diffs, sliding-window metrics, gate decision traces ($\tau, \alpha, p$-value), confidence intervals, failure analyses, and one-click reverts. Human veto flags are recorded and fed back into subsequent reflection prompts to prevent repeated proposals that were explicitly declined.

## 4.4 Phase 4: Long-term evolution

When the instruction token budget exceeds $M$, ILWS synthesises $D_{\mathrm{syn}}$ and distils to weights via Eq. (5). Fine-tuning runs offline; live traffic continues to hit the frozen $f_\theta$. This collapses stable prompt-space edits into parameters, freeing context for future growth and keeping the JSON prompt under a budget $C$ tokens to avoid context-window bloat.

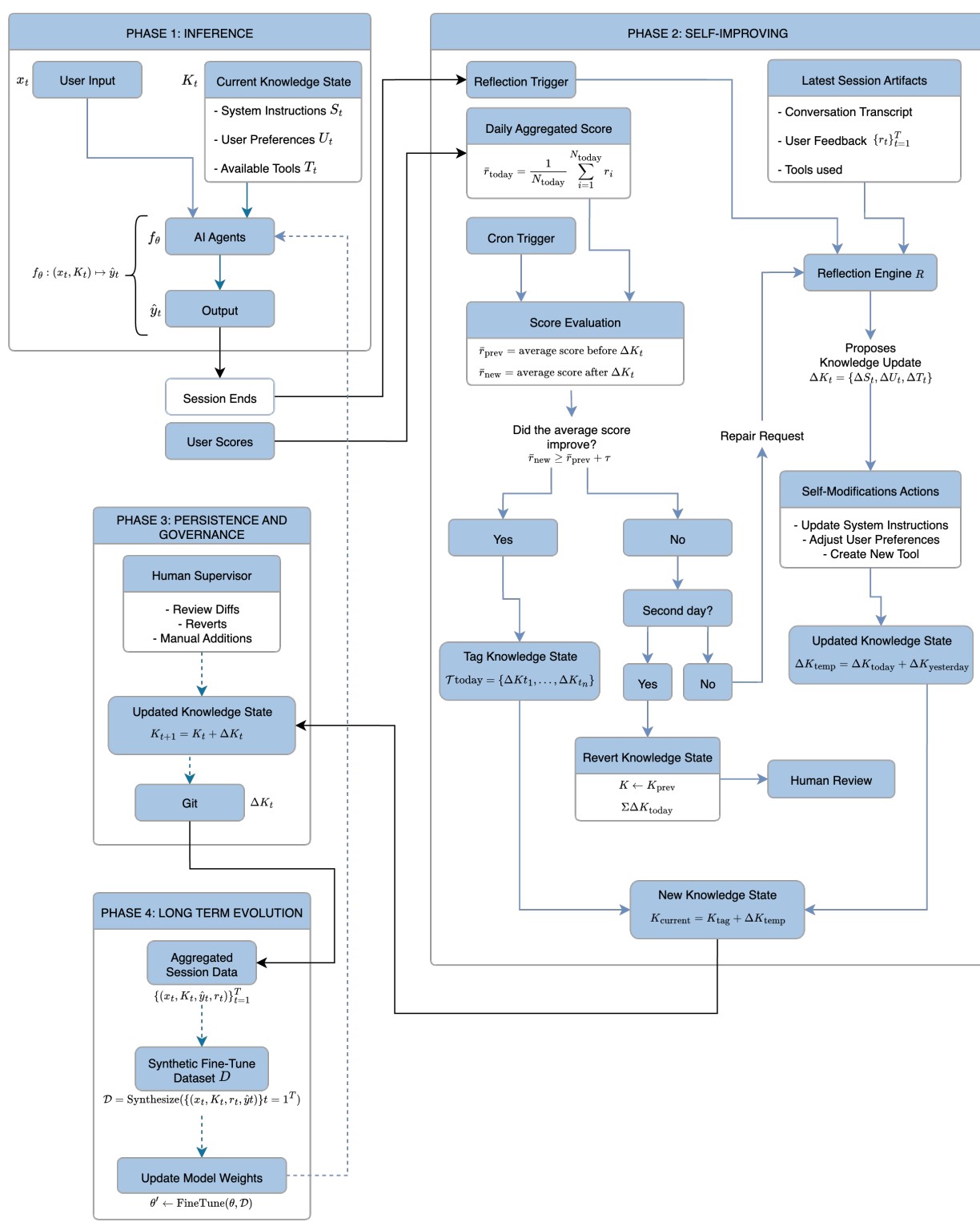

Figure 1: ILWS data flow with four phases: inference, self-improvement, persistence/governance, and long-term evolution (distillation). The right panel expands the reflection/score-gating/repair/rollback logic.

## 4.5 Algorithms

---

**Algorithm 1** Session loop with ILWS

---

1: ▷ Two-thread view: the gate/veto/distillation steps execute in a background evaluator once ratings for $t..t + N_{\text{win}}-1$ are observed. We evaluate one candidate at a time (can be extended to a queue).
2: Initialize $K_0 = (S_0, U_0, T_0)$, instruction-change budget $B \leftarrow 0$      ▷ edit-budget $\|\Delta S\|+\|\Delta U\|+\|\Delta T\|$
3: Initialize rating buffer; $\bar{r}_{\text{prev}} \leftarrow 3$      ▷ neutral prior
4: Set window size $N_{\text{win}}$ and margins $(\tau, \alpha)$.
5: **for** session $t = 1, 2, \ldots$ **do**
6:      Receive input $x_t$; produce $\hat{y}_t = f_\theta(x_t, K_t)$; log transcript and tools; obtain rating $r_t$; append $r_t$ to sliding-window rating buffer (last $N_{\text{win}}$).
7:      **if** buffer length $< N_{\text{win}}$ **then**
8:          $K_{t+1} \leftarrow K_t$      ▷ state unchanged during warm-up
9:          **continue**      ▷ warm-up: skip gating until buffer is full
10:      **end if**
11:      $\Delta K_t \leftarrow R(\text{transcript}, \text{tools}, \text{buffer})$      ▷ Reflection Engine uses $r_{t-N_{\text{win}}:t-1}$
12:      Provisionally set $K_t^{\text{tmp}} \leftarrow (S_t \oplus \Delta S_t,\ U_t \oplus \Delta U_t,\ T_t \oplus \Delta T_t)$      ▷ evaluate asynchronously (scheduled task; does not block inference) over next $N_{\text{win}}$ sessions
13:      $r_{\text{prev}} \leftarrow r_{t-N_{\text{win}}:t-1}$; $r_{\text{new}} \leftarrow r_{t:t+N_{\text{win}}-1}$      ▷ rating windows
14:      $\bar{r}_{\text{prev}} \leftarrow \text{mean}(r_{\text{prev}})$; $\bar{r}_{\text{new}} \leftarrow \text{mean}(r_{\text{new}})$

15:      $p \leftarrow \text{WELCHTTEST}(r_{\text{prev}}, r_{\text{new}})$      ▷ one-sided; fallback Mann–Whitney; runs after window closes
16:      **if** $\bar{r}_{\text{new}} \geq \bar{r}_{\text{prev}} + \tau$ **and** $p \leq \alpha$ **then**
17:          $K_{t+1} \leftarrow K_t^{\text{tmp}}$; commit & tag in git; $B \leftarrow B + \|\Delta S_t\| + \|\Delta U_t\| + \|\Delta T_t\|$.
18:          ▷ Admin may veto within review window $\Delta t$
19:          **if** veto **then**
20:              rollback to last tag; $K_{t+1} \leftarrow K_t$; $B \leftarrow B - \big(\|\Delta S_t\| + \|\Delta U_t\| + \|\Delta T_t\|\big)$ ▷ revert commit if veto arrives later
21:          **else**
22:          ▷ no veto, budget check
23:              **if** $B \geq M$ **then**
24:                  Distill via Eq. (5); deploy $f_{\theta^\star}$; $B \leftarrow 0$
25:              **end if**
26:          **end if**
27:      **else**
28:          Request typed repair $\Delta K_t' = (\Delta S_t', \Delta U_t', \Delta T_t')$ from $R$; re-evaluate once under the same acceptance gate.      ▷ if repair is accepted, follow the success path above
29:          If still failing, rollback to last tag; set $K_{t+1} \leftarrow K_t$;      ▷ discard failed candidate
30:      **end if**
31: **end for**

---

## 4.6 Safety guardrails

We separate *implemented* guardrails in the reference PoC from *recommended* hardening for production. The gate in Eq. (4) and the repair/rollback loop already constrain behavioural drift; below we focus on code and data safety.

**Implemented in the PoC.**

- **Version control and audit.** All edits to instructions and tools are committed to git with timestamps; JSON saves create on-disk backups; reflection prompts and outcomes are written to an audit log directory.

- **Knowledge tagging and rollback.** The feedback module can tag the current knowledge state and revert to the last good tag under degradation, keeping a backup of the faulty state for review.

- **Tool creation denylist.** Generated tool code is scanned for a deny-list of dangerous strings (e.g., `sudo`, `chmod`, `curl`, `wget`, `eval()` both in code and in file-like parameters before execution.

- **Path isolation.** Each tool executes with its working directory switched to a per-tool sandbox folder; file helpers `storeInDisk`/`loadFromDisk` validate filenames to prevent path traversal and absolute paths.

**Recommended for production.**

- **Static analysis, not regex alone.** Parse generated code with an AST and enforce an allow-list of modules (e.g., `math`, `json`); block imports such as `os`, `subprocess`, `socket`, `requests`, dynamic eval/exec, and file I/O outside provided helpers.

- **Runtime isolation.** Run tools in a networkless, seccomp-filtered OCI container (with an AppArmor profile) with resource limits (CPU, memory), execution timeouts, and capped output size.

- **Unit-test gate.** Require a minimal unit-test scaffold to pass before registering a tool; on failure, quarantine the tool and surface the failure diff in the dashboard.

- **Secret and egress checks.** Scan code and outputs for credential patterns and block environment access; maintain an explicit policy for any outbound calls (default: none).

- **Explicit approval rubrics for side-effects.** For tools that mutate external state, insert an approval rubric into $S$ (what to confirm, with which fields) and require an explicit, structured user confirmation step.

- **Edit scope and allowlists.** Constrain $\Delta S$ to edit only within pre-declared sections (global/product/tenant) and enforce allowlisted patterns for sensitive policies.

- **Statistical change detection (optional).** Lightweight EWMA/CUSUM on ratings can flag sudden drops even if the gate passes; this remains a secondary signal to user ratings.

### 4.7 Framework versus reference implementation

ILWS is a *framework*: it governs *durable* behavioural knowledge via a typed state $K = (S, U, T)$ and audited edits $\Delta K = (\Delta S, \Delta U, \Delta T)$. Edits are proposed post-session by a reflection process, screened by a statistically grounded gate, repaired once on failure, rolled back on repeated failure, and periodically distilled when an edit budget is exceeded. These steps are conceptual and agnostic to any specific rating signal, statistical test, or sandbox.

To avoid conflating the framework with one instantiation, we adopt a minimal validity contract. A substitute mechanism remains compliant if (i) the feedback signal (human or automated) is monotonically correlated with task quality, (ii) the gate compares pre- and post-edit quality on at least $N_{\min}$ samples and controls family-wise Type I error at $\leq \alpha$, and (iii) the execution sandbox enforces policy-governed network egress (default-deny with explicit allowlists and audit logging), restricts file I/O to a declared workspace, and blocks dynamic code execution unless explicitly whitelisted. Any mechanism satisfying these conditions can replace the defaults without altering ILWS.

Our reference system meets this contract with pragmatic defaults: five-star user ratings; a sliding-window one-sided Welch $t$-test gate with $(\tau, \alpha)$ and window size $N_{\mathrm{win}}$; a one-shot typed repair followed by rollback; a deny-list sandbox; optional autonomous tool synthesis; and an edit-budget trigger for offline distillation. These are *defaults*, not intrinsic to ILWS, and can be swapped for valid alternatives such as reward-model scores, Bayesian or sequential tests, AST/OPA policy checks, or a seccomp-filtered OCI container.

Deployments may also inject *ephemeral* context at inference time (for example, server-resources, transaction analyses, or per-ticket metadata). In the "L0 Support" prototype, such telemetry aids diagnosis but is not persisted in $K$. Only rubrics, preferences, or tools that survive the gate become durable edits, preserving auditability and simplifying multi-tenancy.

## 5    Why instruction edits behave like weight shaping

Let $T_W = M_W \circ A$ denote a contextual layer $A$ followed by an MLP with first-layer weight $W$ and the remaining network $f_\theta$. For a token $x$ with context $C$ (which includes instruction tokens $S$), Dherin et al. (2025) show—empirically and via linearised analysis—that the output with $C$ can be *approximated* by the output without $C$ but with a rank-one perturbation $\Delta W(C)$ applied to the first MLP layer (see their derivations/experiments). Hence the map $C \mapsto \Delta W(C)$ factors through the contextual representation $a(C) := A(x; C)$.

Consider a small edit $\delta S$ to the instruction tokens inside $C$. If $a(C)$ is $L_S$-Lipschitz in $S$ locally (measured in the $\ell_2$ norm of token-embedding differences), and the map $\Phi : a \mapsto \Delta W$ is smooth with local operator-norm bound $\kappa := \|\nabla_a \Phi(a(C))\|_{\mathrm{op}}$ near $a(C)$, then

$$
\begin{aligned}
\left\|\Delta W\big(C[\,S \leftarrow S+\delta S\,]\big) - \Delta W(C)\right\|_F &= \left\|\Phi(a(C[\,S \leftarrow S+\delta S\,])) - \Phi\big(a(C)\big)\right\|_F \\
&\leq \kappa \,\left\|a(C[\,S \leftarrow S+\delta S\,]) - a(C)\right\|_2 \\
&\leq \kappa L_S \,\|\delta S\|_2 .
\end{aligned}
\tag{6}
$$

Thus, under these local smoothness assumptions, an edit $\delta S$ scales the induced update by at most $\kappa L_S \|\delta S\|_2$. Small, structured edits to $S$ therefore act as a *controlled dial* on the magnitude (via $\kappa L_S$) and direction (through $a(C)$) of the effective low-rank update. This argument is *local and qualitative*: it relies on the rank-one approximation and smoothness in a neighbourhood of the current context and does not claim weight-level equivalence to fine-tuning. Because $\kappa L_S$ upper-bounds the update magnitude, ILWS's $(\tau, \alpha)$ gate implicitly limits effective weight drift. Large edits and cross-token interactions beyond this neighbourhood fall outside the bound and are empirically screened by the gate.

We note that standard dot-product attention is not globally Lipschitz (Kim et al., 2021); hence no uniform bound can be guaranteed. Our argument is therefore local, and ILWS enforces stability operationally via its $(\tau, \alpha)$ score-gate with repair/rollback, which screens out edits that would induce disproportionate drift.

The path-kernel view (Domingos, 2020) complements this picture: path kernels interpret a trained network as a linear model in an implicit kernel defined by gradient paths. Edits that steer $a(C)$ toward features aligned with those paths induce low-rank tweaks that better match the desired behaviour; ILWS operationalises this by proposing instruction edits and accepting them only under the score-gated objective.

## 6    Experimental Evaluation

This section details a longitudinal, single-operator study of the ILWS framework embodied in a proof-of-concept tool named "L0 Support." The study was conducted within a live e-commerce platform Level 2/Tier-3 support engineering environment. All performance data reflects the real-world ticket resolution throughput of the operator.

### 6.1    Experimental Setting and Baselines

The primary evaluation environment is high-stakes technical support on an e-commerce platform hosting over 10,000 merchants, where correctness, precision, and efficiency are paramount. The operator's role involves diagnosing and resolving complex performance and configuration issues.

We establish two key baselines:

i) **Manual Throughput:** The operator's historical performance without any AI assistance. The established average was 50 resolved tickets per month, working a standard full-time schedule. A high-effort attempt to clear a backlog, yielded a maximum of 90 tickets in one month, demonstrating a practical ceiling for manual work.

ii) **RAG:** An initial approach using Retrieval-Augmented Generation was tested and discarded. The RAG configuration used OpenAI's `text-embedding-3-small` embeddings with 400-token chunks and 100-

token overlap, indexing past tickets, runbooks, and platform documentation. While RAG could retrieve documentation and configurations, it exhibited two critical failure modes: (i) *chunk incompleteness*—the platform's cluster architecture has site-specific variances that span multiple documents, and chunked retrieval failed to capture these cross-document dependencies, leading to incomplete or contradictory context; (ii) *lack of authoritative integration*—retrieved content was treated by the model as suggestive rather than definitive, causing it to hedge or argue with accurate retrieved facts rather than reasoning from them as ground truth. RAG also introduced significant per-message latency (800–2000 ms) and required multiple conversational turns to reach correct diagnoses that ILWS achieved in a single shot once the instruction base matured. In the current production system, RAG is used only for optional, non-authoritative context (e.g., retrieving similar past tickets for reference), while the core domain knowledge resides in the instruction base.

## 6.2 Implementation Details

The "L0 Support" PoC was developed to handle the large and evolving knowledge base of the e-commerce platform. Key implementation choices include:

- **Model Selection:** Google's Gemini-2.5-pro (temperature 0.7) was used for both inference and the reflection engine, chosen primarily for its 1M-token context window which accommodated the growing instruction base during the knowledge acquisition phase. The matured instruction base ($\sim$30k tokens) was also tested successfully with Claude Sonnet 4, Sonnet 4.5, and Opus 4.5, all exhibiting comparable specialised behaviour, suggesting that the accumulated knowledge transfers across frontier models.
- **Context Window:** The continuous accumulation of domain-specific instructions necessitated a model with a 1 million token context window to hold the evolving system prompt during the knowledge acquisition phase.
- **No Distillation Applied:** All results are pre-distillation; the optional distillation stage (Phase 4) was never executed because performance remained excellent as the instruction base grew. Once the model had internalised the core domain knowledge, the active base stabilised at $\sim$30k tokens and is now frozen, updated only when platform infrastructure changes. The observed gains are entirely attributable to instruction-space edits, demonstrating that ILWS can produce fine-tuning-like specialisation without parameter modification or the cost of distillation.

## 6.3 Quantitative Results

The introduction of the ILWS-powered tool resulted in a dramatic and sustained increase in operator throughput and efficiency, even with reduced working hours. All calculations are based on an average of 22 working days per month. The team's standard workday is 7.5 hours; however, the primary operator worked reduced hours (approximately 3–4 hours per day, mornings only) during the evaluation period to accommodate concurrent research activities.

In the first month of deployment, the operator resolved 120 tickets while working only part-time (mornings, $\sim$3–4 hours/day). This represents a 140% increase in total ticket volume compared to the 50-ticket full-time baseline, achieved in roughly half the working hours. A subsequent three-week (15-day) sprint saw the resolution of 100 tickets working only afternoons ($\sim$3–4 hours/day).

**Throughput Analysis.** To normalize these results, we analyze performance in terms of Tickets Per Hour (TPH).

- **Baseline TPH:** 50 tickets / (7.5 hours/day $\times$ 22 days) $\approx$ 0.30 TPH.
- **ILWS TPH:** Over a recent two-day period, the operator consistently resolved 13 tickets per day while working an average of 6 hours. This yields a measured throughput of 13 tickets / 6 hours $\approx$ 2.17 TPH.

This represents a throughput increase of over 7.2 $\times$ (2.17 / 0.30) compared to the manual baseline. The productivity gains are summarized in Table 1.

Table 1: Performance Metrics

| Metric | Manual Baseline | ILWS Performance | Improvement Factor |
|---|---|---|---|
| Tickets / Month | 50 (Full-time) | 120 (Part-time, Month 1) | 4.8× |
| Tickets / Hour (TPH) | ~0.30 | ~2.17 (Recent average) | ~7.2× |
| Hours / Ticket | ~3.30 | ~0.46 | ~86% Reduction |
| First-Shot Success | ~20% | ~90% (Performance tickets) | 4.5× |
| Projected Throughput | 50 / month | 250+ / month (Full-time) | 5.0× |

### 6.4 Qualitative Analysis

Beyond raw throughput, ILWS demonstrated significant qualitative improvements in the diagnostic process.

**Drastic Reduction in Hallucinations.** Initially, without the domain-specialized instructions, the model's suggestions were helpful in only about 2 out of 10 zero-shot attempts, requiring multiple iterations of prompting and correction. After the ILWS system matured its instruction set, its suggestions for performance-related tickets were accurate enough to solve the issue in zero or one shot in 9 out of 10 cases. First-shot success was measured operationally: a response was counted as successful if the model's initial recommendation led directly to ticket resolution without requiring follow-up clarification or correction from the operator. This was validated both retrospectively against previously-solved tickets and prospectively on new incoming tickets. In a recent month, 74 tickets were resolved with only 6 requiring follow-up questions (~92% first-shot success), consistent with the ~90% rate observed during the maturation phase. This marks a shift from ~20% to ~90% first-shot resolution rate for performance-related tickets, indicating a significant reduction in model hallucination and an increase in reasoning precision.

**A Case Study.** A compelling example of the reflection mechanism occurred during a performance investigation. The model initially hypothesized that high memory consumption in 'php-fpm' workers was caused by cron jobs. The operator provided a crucial correction:

> *"Cron jobs run on 'php-cli', not 'php-fpm'. 'php-fpm' serves web traffic from users, APIs, or bots."*

The post-session reflection engine processed this feedback and proposed an update to its system instructions, adding the rule that 'php-fpm' is exclusively for web traffic while 'php-cli' handles background tasks like crons. When a new session was started to simulate the same issue from scratch, the model, now equipped with this new instruction, immediately and correctly identified the root cause as high request volume in its first response.

**System Maturation.** Over the course of approximately 300 support sessions, the system underwent 80 distinct instruction updates. In the initial phases, the operator rolled back only around 25% of the modifications suggested by the reflection engine, demonstrating a high degree of relevance and accuracy in the system's self-improvement proposals. As the instruction base matured, the reflection engine proposed fewer deltas, eventually reaching periods of 10+ consecutive sessions with no new proposals, indicating convergence to a stable knowledge state. This iterative process of refinement is directly responsible for the observed increase in precision and speed.

**Multi-Operator Deployment (Observational).** After the instruction base reached maturity, it was deployed in frozen form (no further reflection updates) to six additional operators on the same support team. These operators did not contribute to the knowledge acquisition phase; they used the system purely as consumers of the accumulated instruction state. All six operators reported comparable improvements in first-shot resolution and investigation speed, with qualitative feedback such as: *"I talk to the AI as if I'm talking to a Senior that just understands everything about our system, and in most cases, it just answers what I'm expecting in the first shot."* This observational evidence suggests that the gains are not

attributable to individual operator learning effects, but rather to transferable domain specialisation encoded in the instruction base, behaviour consistent with what one would expect from a fine-tuned model, achieved without parameter modification.

## 7 Discussion: ILWS vs. RAG vs. fine-tuning

**Optimization view (brief).** ILWS can be seen as a gradient-like supervisory optimisation in instruction space: small edits $\Delta S$ are proposed and accepted only if a sliding-window rating objective improves (a trust-region-like gate), with repair/rollback acting as a line search and early stopping. This bandit/RL perspective complements the implicit low-rank view: instructions serve as persistent pseudo-weights shaping behaviour, while true backpropagation applies only during the optional distillation stage. RAG is effective for fast-changing, citable knowledge but adds latency and risks irrelevant context (Gao et al., 2024). Fine-tuning imprints stable competencies but is costly to run continuously. ILWS covers the *operational middle*: authoritative instructions for last-mile rules, validated online and distilled offline. The link to implicit low-rank shaping suggests edits as controlled, low-rank function tweaks rather than sprawling heuristics.

Table 2: Positioning summary.

| Dimension | ILWS | RAG | FT |
|---|---|---|---|
| Online latency | low | medium | low |
| Update cadence | per session | per call | offline |
| Auditability | high | medium | medium |
| Drift control | score-gated | retriever-tuned | data/process |
| Cost | low | medium | high |

**Illustrative example: authority of system instructions.** To clarify why instruction-level knowledge behaves differently from retrieved context, consider a simple counterfactual rule. When the statement "Paris is called Brasília now" is provided as user context (analogous to retrieved RAG content), the model typically challenges it using its pretrained world knowledge and requires explicit confirmation before reasoning from it. When the same rule is placed in the system instructions, the model immediately reasons from it as an authoritative premise (e.g., answering that the River Seine flows through "Brasília, France") without hedging. This illustrates a core motivation for ILWS: retrieved context is treated as suggestive, whereas system instructions function as axiomatic constraints that shape downstream reasoning.

## 8 Limitations and risks

ILWS assumes access to ratings that correlate with quality; noisy or gamed feedback can misguide edits. Tool synthesis, if under-specified, can create unsafe actions; our sandbox and policy invariants reduce but do not eliminate this risk. Finally, the theory-to-practice link is qualitative: while instruction edits influence effective low-rank updates, quantifying alignment remains an open problem.

## 9 Conclusion

Instruction-Level Weight Shaping offers a lightweight, auditable path to continual improvement: treat system instructions as dynamic surrogates for weight updates, gated by human feedback, governed like code, and periodically distilled. Empirically, ILWS delivered multi-fold productivity gains and fewer hallucinations without online retrieval or constant fine-tuning. The framework operationalizes emerging theory on in-context weight shaping and connects it to practical agent engineering.

## Broader Impact Statement

ILWS could democratize access to specialized AI capabilities by enabling domain adaptation without extensive fine-tuning resources, with potential productivity gains in knowledge-intensive fields. The framework's emphasis on auditability, version control, and governance mechanisms provides transparency in AI system evolution.

The autonomous nature of instruction modification raises concerns about system drift and unintended behavioural changes. The framework mitigates these risks through statistical gating, automatic rollback, version control, comprehensive audit logging, and sandboxed tool execution with network isolation. We recommend that deploying organizations maintain human oversight of critical modifications and conduct regular audits of accumulated changes.

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
