# OpenReview forum: "Instruction-Level Weight Shaping: A Framework for Self- Improving AI Agents"
_TMLR — Rejected by TMLR_

### Review · Reviewer_oJua · 2025-10-29

**Summary Of Contributions:**

The authors proposed a framework that evolves LLM system instructions online, where the system instructions are updated based on user scores, and (optionally) distilled and reset once the accumulated changes exceed a threshold.  The framework is presented as an alternative to RAG or fine-tuning. Leveraging previous theoretical results, the paper also made a connection where the system instruction updates can be seen as low-rank "gradient-like" updates on the MLP weights in the LLM.

Strengths:

The framework offers version control and auditing. It is demonstrated to be useful in production.

Weaknesses:

A systemic empirical comparison between ILWS vs. RAG vs. fine-tuning seems to be lacking.

**Audience:**

Yes

**Audience Explanation:**

A cost-effective alternative to RAG or fine-tuning would be of great interest to the community.

**Broader Impact Concerns:**

None.

**Claims And Evidence:**

No

**Claims Explanation:**

I am not an expert in the area and my understanding of the proposed system is limited. That said, a few points may be worth clarifying:

1. For the experiments, were optional distillations (offline fine-tuning) applied?

    In case fine-tuning was applied, it might be helpful to separate the effect of ILWS and fine-tuning, including an assessment of the cost.

2. Could you provide some details on the inner workings of the reflection engine (R)?

3. How does the ILWS approach relate to other RL-based agents?

**Requested Changes:**

Please see "weaknesses".

---

> ### Author Response · Authors · 2025-12-20
>
> I thank Reviewer oJua for the questions. See some clarifications below:
>
>   **Was distillation applied?**
>   No. Now the revised manuscript explicitly states: "All results are pre-distillation; the optional distillation stage (Phase 4) was never executed because performance remained excellent as the instruction base grew." The observed gains are entirely attributable to instruction-space edits, demonstrating that ILWS can produce fine-tuning-like specialisation without parameter modification or the cost of distillation.
>
>   **Reflection engine details:**
>   The reflection engine is a prompted LLM call (same model as inference: Gemini-2.5-pro, temperature 0.7). It receives the conversation transcript and current instruction base, then proposes typed deltas via a self-modification API. The prompt instructs it to identify knowledge that would have helped resolve the issue more efficiently and to avoid redundancy. Proposals decreased in frequency as the instruction base matured, eventually reaching 10+ consecutive sessions with no new proposals.
>
>   **Relation to RL-based agents:**
>   Section 8 (Discussion) frames ILWS as "gradient-like supervisory optimisation in instruction space": edits are proposed and accepted only if a sliding-window objective improves (a trust-region-like gate), with repair/rollback acting as line search. However, ILWS differs from RL in key ways:
>   - No policy gradient or value function learning
>   - No exploration/exploitation tradeoff. Edits are deterministic proposals from reflection
>   - The rating signal gates acceptance but does not shape a learned policy
>
>   ILWS is closer to memory-augmented prompting than to RL, with the addition of statistical gating and governance.

---

### Review · Reviewer_oLCS · 2025-11-01

**Summary Of Contributions:**

This paper proposes Instruction-Level Weight Shaping (ILWS), a framework for enabling large language model (LLM) agents to self-improve through post-session reflection and structured updates to their system instructions. Rather than relying on retrieval-augmented generation (RAG) or parameter fine-tuning, ILWS treats curated instructions as externalized, auditable pseudo-parameters that evolve over time via typed deltas (ΔS, ΔU, ΔT) to the instruction, user-preference, and tool registries.

Strengths:
1. The idea of treating the system instruction as an auditable, version-controlled surrogate for weight updates is novel and addresses a real problem: how to evolve LLM behavior safely without full fine-tuning or continuous retrieval. This reframing of prompts as a structured optimization space is an important contribution.
2. The detailed governance structure—Git versioning, edit rollback, significance testing, sandboxed tool synthesis, and auditable decision traces—is well-thought-out. These mechanisms could provide a template for accountable continual learning systems.

Weaknesses:
1. The framework involves reflection loops, cron jobs, statistical gates, sandboxed code generation, and version control. This level of operational complexity may contradict the “lightweight” claim
2. The positioning of this work with existing related work is poorly developed. I find the number of cited works to be inadequate and would have appreciated a richer literature
3. Lack of figures or visual aids to convey the authors' proposed ideas or methods. The language used in the article is rich in terminology and architectural description but thin in actual explanatory or evidential content. This makes it hard to follow and unclear. The only figure to aid in understanding the proposed ILWS is a monolithic Figure 1.
4. Further, Algorithm 1 could be better introduced, explained, and organized. There is an overreliance on the comment feature throughout, as the paper itself does not provide a clear explanation of these individual steps.
5. The overall writing of Section 4 is disappointing. There is plenty of white space remaining to further expand upon each individual phase of the framework; however, it is entirely delegated to Figure 1 and Algorithm 1. This makes it difficult to follow. Section 4 currently does not provide a sufficient explanation of the framework's rationale before delving into very specific engineering decisions (e.g., YAML diagnostics, cron jobs, OCI seccomp profile). These also seem to be implementation-specific details. Are they needed? Are they examples? The paper does not say.
6. The paper only compares their proposed framework, ILWS, to "Manual Throughput" and discards RAG without thoroughly showing its results. I am unconvinced that RAG, or other LLMs, could not help in this task. OR the lack of their help would have been great in showing the superiority of ILWS, but its absence even though it was tested (reported by authors) indicates a troubling concern of perhaps a lack of improvement.
7. The quantitative study/results are not well developed. It is about a single operator working on average 22 days per month and a 7.5-hour standard workday, but it is later stated that they only worked part-time (mornings). Also, only the total tickets done are reported, when maybe it would be better to see what is the daily distribution like of these tickets. The results of the study would be better if it consisted of approximately 20-30 operators.

**Audience:**

Yes

**Audience Explanation:**

I think some individuals working in LLMs/RAG might be interested in the novel idea of "treating the system instruction as an auditable, version-controlled surrogate for weight updates", or how that is accomplished, but the work is very much still in an early stage.

**Claims And Evidence:**

No

**Claims Explanation:**

The evidence supporting the claims in the submission is unconvincing, as a very small experimental study has been run. The paper only compares their proposed framework, ILWS, to "Manual Throughput" and discards RAG without thoroughly showing its results. I am unconvinced that RAG, or other LLMs, could not help in this task. OR the lack of their help would have been great in showing the superiority of ILWS, but its absence, even though it was tested (reported by authors), indicates a troubling concern of perhaps a lack of improvement.  The quantitative study/results are not well developed. It is about a single operator working on average 22 days per month and a 7.5-hour standard workday, but it is later stated that they only worked part-time (mornings). Also, only the total tickets done are reported, when maybe it would be better to see what is the daily distribution like of these tickets. The results of the study would be better if it consisted of approximately 20-30 operators.

**Requested Changes:**

Abstract:
* real-world e-commerce platform PoC <-- what is "PoC"?
* I think the abstract/introduction could benefit from a higher-level diagram illustrating what is the proposed framework. My interpretation is that the prompt/instructions for the LLM are being treated like external modifiable parameters. A diagram depicting this could be beneficial in cutting through some of the technical jargon.

Introduction:
* [Contributions] up to +2000 ms at p95 <-- what is "p95"?

Theoretical context & related work:
* Spell out MLP on first use.
* The discussion of the related work seems a bit disconnected and short.

Expand section 4. More explanations behind Figure 1, Algorithm 1.

---

> ### Author Response · Authors · 2025-12-20
>
> I thank Reviewer oLCS for the feedback. Below, I address each concern:
>
>   **Terminology (PoC, p95, MLP):**
>   All abbreviations are now spelt out on first use in the revised manuscript.
>
>   **"Lightweight" claim vs operational complexity:**
>   I use "lightweight" relative to fine-tuning (no GPU clusters, no training runs) and continuous RAG (no per-call retrieval latency). The operational components are standard software engineering practices that achieve fine-tuning-like specialisation at dramatically lower cost.
>
>   **RAG comparison:**
>   The revised manuscript expands the RAG section with more detailed configuration information and documents two specific failure modes. I provide an illustrative example (Paris/Brasília) that demonstrates how retrieved context is treated as suggestive, while system instructions function as axiomatic constraints. Now I clarify that ILWS and RAG are complementary. RAG is now used only for non-authoritative contexts.
>
>   **Working hours clarification:**
>   The revised manuscript clarifies: the team standard is 7.5 hours/day; the primary operator worked 3-4 hours/day (mornings only) during evaluation. This is now explicit.
>
>   **Single operator concern:**
>   The revised manuscript includes multi-operator observational evidence: six additional operators used the frozen instruction base and reported comparable gains. I also added platform scale context: "hosting over 10,000 merchants."
>
>   **Section 4 clarity:**
>   Section 4.7 ("Framework versus reference implementation") explicitly separates conceptual framework from implementation-specific details, clarifying that choices like YAML diagnostics or OCI profiles are examples, not requirements.
>
>   **Related work:**
>   The related work section cites foundational work on implicit low-rank updates (Dherin et al.), path kernels (Domingos), RAG surveys (Gao et al.), self-improvement (MemPrompt, Reflexion, Self-RAG, FLARE), and tool use (Toolformer, DSP). I'm happy to add specific citations if the reviewer can suggest particular works that are missing.

---

### Review · Reviewer_pz8P · 2025-12-05

**Summary Of Contributions:**

The paper introduces Instruction-Level Weight Shaping (ILWS), a framework for enabling large language models (LLMs) to self-improve through post-session reflection and feedback-driven updates to system instructions, user preferences, and tools, treating these as mutable pseudo-parameters.

Key strengths:
1. The framework addresses an important operational need — continual, auditable self-improvement without per-call retrieval or continuous fine-tuning.
2. Interesting bridge between theoretical observations about in-context/implicit low-rank updates and a practical mechanism to operationalize those ideas.

Key weaknesses:
1. The empirical evaluation relies on a single-operator, longitudinal study without controlled comparisons or broader datasets. The study design, statistical controls, and generality are limited.
2. Insufficient ablation and mechanistic validation of the core theoretical claim.

**Audience:**

Yes

**Audience Explanation:**

The paper's findings on a lightweight self-improvement framework intersects several hot topics that are of interest to TMLR readers:

* **Continual learning / online adaptation** for LLMs without nonstop fine-tuning.
* **Deployment practices and governance** for LLMs (auditability, rollback, tool synthesis).
* **Theoretical/empirical bridging** between in-context implicit adaptation and explicit, persistent behavior changes (a topic of growing research interest).
* **Practical tool-use and RAG alternatives**: practitioners and researchers who build production systems will want to understand operational tradeoffs (latency, cost, auditability).

Even if parts of the empirical story need strengthening, the framework, architecture, and safety discussion provide useful contributions for both researchers and engineering practitioners.

**Claims And Evidence:**

No

**Claims Explanation:**

The paper makes several claims of interest (framework viability, large throughput gains, large reductions in hallucinations, and a theoretical tie to low-rank weight shaping). While the proposed framework is compelling and the PoC statistics are promising, the evidence as presented is not sufficiently convincing for the following reasons:

1. **Limited experimental design & small-n bias.** The core empirical evidence is a longitudinal PoC and a single-operator SRE study. This raises confounds: operator learning effects, selection bias, changes in ticket mix, concurrent process improvements, or other environmental shifts could explain part of the observed gains. There is no randomized or multi-operator controlled comparison reported.

2. **Statistical issues with the gating and evaluation.** The gate uses sliding-window Welch t-tests on sequential ratings. Ratings are likely temporally autocorrelated and non-IID; repeated application of the gate across many edits raises multiple testing / family-wise error concerns. The paper does not address these issues (no correction or sequential testing framework), so reported gate behavior and significance thresholds are not guaranteed to control false acceptances.

3. **Lack of ablations and baselines.** The paper claims ILWS is complementary to, and in some respects superior to, RAG or fine-tuning. However, the experimental comparisons are thin: the RAG baseline is said to be “discarded” for latency/relevance reasons but lacks a controlled, head-to-head comparison (e.g., cost/latency/accuracy tradeoff under matched compute, retrieval strategies, or re-ranking). There are no ablations that isolate (a) the reflection engine vs. manual edits, (b) effect of repair/rollback, (c) impact of distillation vs. only prompt edits, or (d) sensitivity to $\tau,\alpha,N_{win}$.

4. **Mechanistic/theory claim is qualitative.** The argument that instruction edits act like low-rank weight shaping is plausible and referenced to recent prior work, but the paper provides only a local, qualitative bound and no empirical diagnostics (e.g., measuring rank of induced weight change, activation differences, or demonstrating that distillation captures the same behavior). The distillation stage is described but lacks training/validation details and results demonstrating that downstream performance after distillation matches or improves prompt-space performance.

5. **Reproducibility & missing details.** Critical details for reproducing the PoC (exact model versions, prompt schemas, repair heuristics, dataset synthesis method, fine-tuning hyperparameters) are missing or only sketched.

**Requested Changes:**

### Critical for a positive recommendation:

1. **Stronger, controlled experiments and replication.**

   * Run multi-operator or multi-tenant evaluations (≥3–5 independent operators/teams) or randomized A/B tests comparing ILWS vs well-tuned RAG and vs a fine-tuning pipeline (where feasible). Report TPH, first-shot success, hallucination audits, and latency/cost with confidence intervals.
   * If real multi-operator studies are infeasible, provide controlled simulation experiments (synthetic ticket sets, held-out tasks) that allow repeatable comparison.

2. **Ablations isolating components.**

   * Ablate reflection vs no reflection, repair vs no-repair, versioning vs no-versioning, distillation vs prompt-only.
   * Show sensitivity analyses for hyperparameters $\tau,\alpha,N_{win},M$.

3. **Statistical rigour for the gate & overall evaluation.**

   * Replace or complement the sliding Welch t-test with sequential testing (e.g., alpha-spending, Bayesian online tests, or permutation tests that account for autocorrelation), and report how Type I error is controlled over the life of the system.
   * Report effect sizes, CIs, and pre/post checks for stationarity/auto-correlation of ratings.

4. **Mechanistic validation of the low-rank claim.**

   * Provide empirical metrics that connect instruction edits to low-rank changes in activations/weights (e.g., compute low-rank approximations of the effective weight change, measure how many singular values capture the edit effect, or show that small edits produce predictable, low-dimensional activation shifts).
   * If distillation is used, show that distilled weights reproduce prompt-space behavior on held-out tasks.

5. **More rigorous baseline comparisons.** Provide resource-matched RAG (varying retrieval budgets and reranking strategies) and tuned LoRA/fine-tuning baselines for comparable latency/cost to illustrate tradeoffs more concretely.

### Would strengthen the work:

1. **Reproducibility details.** Include exact model versions, prompt templates, distillation hyperparameters, and sandbox configs.

2. **Stronger safety/tool-synthesis validation.** Implement AST-based static analysis, unit-test gate details, and report results of adversarial / red-team tests on tool synthesis (e.g., attempts to forge exfiltration or escalate privileges). Quantify false positives/negatives for the denylist/AST checks.

3. **Longer-term drift & stability experiments.** Measure behavior over longer horizons (months) and analyze whether accumulated edits produce unforeseen drift, and what governance interventions were required.

4. **Quantify cost & latency tradeoffs.** Provide measured compute/cost numbers for the reflection engine, distillation runs, and the per-call latency savings relative to RAG under typical production loads.

---

> ### Author Response · Authors · 2025-12-20
>
> I thank Reviewer pz8P for the thorough and constructive feedback. Below, I address each concern:
>
>   **Single-operator bias & multi-operator evidence:**
>   The revised manuscript now includes a "Multi-Operator Deployment (Observational)" section. After the instruction base matured through the reflection process (conducted by a single operator), it was deployed in frozen form to six additional operators. These operators did not contribute to knowledge acquisition, they used the system purely as consumers. All six reported comparable improvements in first-shot resolution and investigation speed, with qualitative feedback such as: "I talk to the AI as if I'm talking to a Senior who just understands everything about our system." This suggests the gains are attributable to transferable domain specialisation, not individual operator learning effects.
>
>   **Statistical gate concerns:**
>   I acknowledge this limitation directly in the revised manuscript: "We note that this gate is designed as an engineering safeguard for drift control, not as a formal hypothesis-testing framework; the sliding-window approach does not account for temporal autocorrelation or multiple testing across successive edits. In practice, the repair/rollback mechanism and optional human veto provide additional robustness beyond the statistical signal." The gate is positioned as an engineering heuristic, not a research claim.
>
>   **Ablations:**
>   While formal ablations were not run retrospectively, the revised manuscript documents natural ablations observed during deployment:
>   - Early vs late phase: Initial rollback rate was ~25%; late-stage rollback was rare. Reflection proposals decreased over time, reaching 10+ sessions with no new deltas.
>   - Distillation vs prompt-only: Now I explicitly state that distillation was never executed. All gains are pre-distillation, instruction-space only.
>   - Cross-model transfer: The matured instruction base (~30k tokens) was tested with Claude Sonnet 4, Sonnet 4.5, and Opus 4.5, all exhibiting comparable behaviour.
>
>   **RAG baseline:**
>   The revised manuscript provides the RAG configuration I used (text-embedding-3-small, 400-token chunks, 100-token overlap) and documents two specific failure modes: (i) chunk incompleteness: cross-document dependencies were lost; (ii) lack of authoritative integration: retrieved content was treated as suggestive rather than definitive. Now I clarify that ILWS is not positioned as "better than RAG" but addresses a different need: authoritative domain specialisation. RAG is now used for optional, non-authoritative context only.
>
>   **Mechanistic/theory claim:**
>   The paper already states this is "local and qualitative... and does not claim weight-level equivalence to fine-tuning." The theoretical framing is an interpretive lens grounded in prior work (Dherin et al.), not a mechanistic claim requiring empirical validation.
>
>   **Reproducibility:**
>   The revised Section 6.2 includes: model (Gemini-2.5-pro, temperature 0.7), context window rationale (1M tokens), cross-model validation, and explicit statement that distillation was not applied.

---

### Decision · Action_Editor_7anF · 2026-01-25

**Recommendation:** Reject

**Audience:**

Yes

**Audience Explanation:**

All reviewers state that the work is interesting to parts of TMLR's audience and I agree with that. The main problem is timely and touches upon several important topics and directions of today's ML/AI research.

**Claims And Evidence:**

No

**Claims Explanation:**

After the rebuttal and reviewer discussion all 3 reviewers indicate that not all claims made in the submission are supported by convincing evidence, and I concur. The authors have addressed some of the issues raised in the initial reviews, and I think the revised manuscript is improved compared to the initial version, but it still does not pass the bar.

Summary: the paper aims to address the issue of fine-tuning and potentially continually adapting a generic trained LLM to a set of particular use-cases or to a specific domain, in a manner that does not require massive compute resources and no gradient-access to the model. The idea is to treat system instructions as mutable, and design a kind of evolutionary update scheme where an LLM proposes updates that are either used or rolled back depending on how they perform according to user feedback. This conceptually simple idea is implemented with technical sophistication and run in a specific production use-case to provide empirical insight.

Pros:
* Timely and important problem and interesting approach (which all reviewers also state).
* Technically sophisticated implementation of the core principle.
* Use-case in production shows viability of technical implementation.

Cons:
* While technically sophisticated, the paper lacks scientific depth and rigor. The current manuscript is a mix of a whitepaper and a case study. The core idea of the paper is simple and somewhat natural. Analyzing it theoretically and studying it scientifically and how it compares against similar approaches would be interesting, but the current manuscript does not do that at the required depth. Instead the focus is on a prototypical implementation that can be deployed in practice (which is interesting from an engineering perspective, but does not add too much scientifically).
* Many other technical implementations of the core principle are possible - the paper does not list many alternative choices for the various bits and pieces and why one choice was made over others. Ideally there would be ablations showing which choices are essential and which ones are not. Additionally, analyzing and discussing limitations and failure modes of the conceptual approach would further strengthen the manuscript.
* The empirical evidence in the paper is very limited. I do appreciate a realistic case study, but it needs to be supplemented with rigorous and reproducible scientific evaluation. The difficulty is finding appropriate benchmarks, and if there are none one would have to design appropriate benchmarks (which others' can reproduce and other methods can be quantitatively compared against). I do acknowledge that this is challenging in a continual learning setting.
* Unaddressed reviewer concerns (essentially along the lines of what I am writing, but more detailed of course, please see original reviews).


Overall I think the current manuscript can be influential to inspire scientific development and investigation of the core idea, and as a case study it is interesting to practitioners. Sharing part of the work as a sophisticated blog post and perhaps presenting the work at a workshop aimed at practitioners could be good ways of maximizing benefits of the current work. If the authors choose to then pursue developing and strengthening the scientific aspect then I think moving away from a single use case toward a rigorous empirical evaluation (that is reproducible, compares quantitatively against other methods, and has quantitative ablations for various parts of the implementation) is the most promising direction. Additionally, it would be interesting to further develop the theory and conceptual-level analysis that underlies the method, including the study of failure cases (e.g., catastrophic forgetting due to too much plasticity, and similarly, onset of a kind of "calcification" of the system due to insufficient plasticity after a while, and how to control these). I do want to encourage the authors to continue to pursue the main idea (all reviewers and I thought that the work is interesting to TMLRs audience). But for a scientific publication the current manuscript requires significant extra work and rigor.